# A Novel Hyperspectral Method to Detect Moldy Core in Apple Fruits

**DOI:** 10.3390/s22124479

**Published:** 2022-06-14

**Authors:** Andrea Genangeli, Giorgio Allasia, Marco Bindi, Claudio Cantini, Alice Cavaliere, Lorenzo Genesio, Giovanni Giannotta, Franco Miglietta, Beniamino Gioli

**Affiliations:** 1Department of Agronomy and Land Management, University of Florence, P.le delle Cascine 18, 50144 Florence, Italy; andrea.genangeli@unifi.it (A.G.); marco.bindi@unifi.it (M.B.); 2Gruppo FOS, Via Enrico Melen, 77/ed.A, 16152 Genova, Italy; giorgio.allasia@fos.it (G.A.); giovanni.giannotta@fos.it (G.G.); 3Institute of Bioeconomy (IBE), National Research Council (CNR), Via Caproni 8, 50145 Florence, Italy; claudio.cantini@ibe.cnr.it (C.C.); alice.cavaliere@ibe.cnr.it (A.C.); lorenzo.genesio@ibe.cnr.it (L.G.); franco.miglietta@ibe.cnr.it (F.M.)

**Keywords:** moldy core, internal browning, hyperspectral classification, near-infrared spectroscopy, ANN binary classification

## Abstract

An innovative low-cost device based on hyperspectral spectroscopy in the near infrared (NIR) spectral region is proposed for the non-invasive detection of moldy core (MC) in apples. The system, based on light collection by an integrating sphere, was tested on 70 apples cultivar (cv) Golden Delicious infected by *Alternaria alternata*, one of the main pathogens responsible for MC disease. Apples were sampled in vertical and horizontal positions during five measurement rounds in 13 days’ time, and 700 spectral signatures were collected. Spectral correlation together with transmittance temporal patterns and ANOVA showed that the spectral region from 863.38 to 877.69 nm was most linked to MC presence. Then, two binary classification models based on Artificial Neural Network Pattern Recognition (ANN-AP) and Bagging Classifier (BC) with decision trees were developed, revealing a better detection capability by ANN-AP, especially in the early stage of infection, where the predictive accuracy was 100% at round 1 and 97.15% at round 2. In subsequent rounds, the classification results were similar in ANN-AP and BC models. The system proposed surpassed previous MC detection methods, needing only one measurement per fruit, while further research is needed to extend it to different cultivars or fruits.

## 1. Introduction

The apples production and post-harvest industry is among the largest fruit markets at the global scale, given the large diffusion and consumptions of apple fruits [1]. Final consumers are showing an increasing attention towards food quality and sustainability of food supply chains, aiming to consume products with low environmental impact and homogeneous organoleptic characteristics without internal or external alterations [2]. Satisfying these requirements represents one of the main objectives for farmers and food-companies involved in the production and marketing of apples [3,4,5]. One of the most relevant causes of quality loss is represented by the internal browning in apple post-harvest phases caused by *Alternaria* sp. (Asp), a ubiquitous genus fungorum widely present in all apple-growing areas.

The pathology caused by Asp is called mold core, or simply moldy core (MC) and initiates and produces its damage effects in the interior part of the fruits [6,7]. Previous studies have shown that the principal MC susceptible apple cultivars (cv) are represented by Fuji, Red Delicious, and Granny Smith, but other significant varieties, such as Golden Delicious, can still present relevant internal damages by MC [8]. Indeed, apple cv Golden Delicious plays a fundamental role in the world apple industry, especially in Italy, where it is the most widely cultivated apple cv, with a total production volume of 858,423 tons [9,10]. The damage by MC is an internal injury, with dry-brown areas developed in the inner part of the fruit. The fruit does not present any external sign of damage, making MC detection with classical non-invasive methods very challenging or practically impossible [11].

Therefore, the development of non-invasive analytical methods to detect MC damages throughout the production process, especially in post-harvest phases deserves great attention from producers.

Novel non-invasive approaches deriving from different technological sectors have been successfully tested and applied in recent years to detect internal injury or internal browning in apples, such as time-frequency images of vibro-acoustic signals [12,13]; magnetic resonance techniques [14,15] and X-ray analysis [16,17]. Thereby, these techniques deliver a meaningful occasion to explore innovative analysis methods capable to detect qualitative parameters in apples, but at the same time they show limitations due to their cost, size of equipment, and operating time. Another innovative and promising solution with non-invasive methods for quality control in apple production has derived from recent developments by spectroscopy application from field to post-harvest phases [18,19]. Especially, the exploitation of light properties around the near infrared (NIR) region of the electromagnetic spectrum has captured the interest of researchers and industry in recent years, since it provides a valid alternative compared to invasive analysis methods [20,21]. Near infrared spectroscopy (NIRS) technics are based on the collection of spectral information such as absorption (ABS), reflectance (REF), and transmittance (TR) of electromagnetic signals in the spectral region spanning from 700 to 1200 nm [22].

REF based methods measure the reflected spectral signature under controlled illumination conditions and is typically used to retrieve parameters or compounds that are present on the fruit surface. REF was successfully used in several different applications, e.g., real-time quantification of biophysical and biochemical parameters through non-destructive method in citrus [23] and detection of oil palm maturity in bunches of fruits [24]. Techniques based on light reflectance in VIS/NIR have been successfully applied in apples quality control to detect a wide number of biophysical and biochemical parameters, such as external decay in apples [25], degrees brix [26], postharvest storage periods [27], and chlorophyll content [28]. ABS/TR techniques are based on generating a convenient light source at one side of the fruit, which propagates across the interior of the fruit, being finally collected at a convenient escape sensing surface. In recent years, the improvement of efficient machine learning techniques (MLT) has permitted the development of innovative analysis models in different application areas, e.g., engineering science, modeling in geology, and data reduction [29,30,31]. Moreover, MLT associated with TR analysis was successfully used in developing calibration models to quantify brix and pol at various stages of an industrial sugar production process [32]; therefore, the application of these methods is a relatively new field in the agricultural post-harvest science. TR and MLT were applied sporadically on apples and other commercial fruits. Nevertheless, such methods were applied sporadically on apples while they were already successfully tested in different commercial fruits; e.g., TR techniques and convolutional neural network (CNN) were successfully applied in blueberry internal damage detection with a classification accuracy over 80% [33]. Moreover, TR techniques represent an important analysis tool in high-income fruits, e.g., the grade of ripeness in nectarine (Prunus Persica) was evaluated with an accuracy of 88% by TR analysis in combination with Partial Last Square Regression (PLS-R) [34].

In the last few years, interest for MC detection by NIRS has increased in research activities and some different NIRS applications have been proposed. The main studies concerning the assessment of the capability to detect MC presence by VIS/NIR transmittance spectra retrieval were conducted mainly in cv Fuji. Zhaoyong et al. [35] verified MC presence in cv Fuji through an acquisition method based on multiple measurements per fruit; results obtained through the application of classifications algorithms based on back propagation artificial neural network (BP-NN) and support vector machine (SVM) have shown a classification accuracy of infected and healthy apples larger than 83%. Similarly, Shenderey et al. [36] obtained a classification accuracy of healthy and infected apples larger than 90% by PLS-R in cv Fuji with a moldy area larger than 5%. Tian et al. [37] investigated the relationship between the orientation of fruit trough light source position, achieving best results with fruit stem-calyx axis horizontal and perpendicular to transmission belt in apples. These studies highlighted the high complexity of transmittance-based hyperspectral NIR measurements, which are strongly affected by fruit and illumination geometry and by cv specific traits and chromatic characters that strongly affect the measured spectral signatures. Results obtained on a specific cv are, therefore, likely not transposable to different ones.

The objectives of this work were:(i)To develop and validate an innovative and low-cost application of NIRS to detect and monitor MC presence and growth in cv Golden Delicious through a novel measurement system based on a light source—light transmission—light collection architecture. An integrating sphere (IS) with homogeneous light reflectance proprieties [38] was adopted to compensate the geometrical variability in each fruit and toward the illumination geometry, and a low-cost VIS-NIR commercial spectroradiometer was used to measure the transmitted radiance inside the integrating sphere.(ii)To develop spectral based algorithms capable of detecting the MC and classifying the fruits in a binary classification framework (e.g., classifying a fruit as healthy or moldy), based on several state of the art machine learning techniques: pattern recognition neural networks (ANN-AP), Logistic Regression (LR), Linear Support Vector Classification (SVC), Random Forest (RF), Naive Bayes (NB), K-Nearest Neighbor (KNN), and Bagging Classifier based on Decision tree (BC).(iii)To assess the temporal performance of the detection algorithms, i.e., to assess the amount of time after the inoculus at which it becomes detectable.(iv)To assess the sensitivity of the algorithms, i.e., the minimum amounts of infected tissues that can be detected.(v)To determine the most important spectral bands responsible for the MC detection, and the minimum number of bands that can be used to further develop low-cost-multispectral rather than hyperspectral detectors.

Finally, the technological and industrial implications of the proposed sensing technology are discussed.

## 2. Materials and Methods

### 2.1. Instrument Setup

The prototype measurement system to detected MC in apples presented here is called Apple Light Transmittance System (ALT-S) and is shown in Figure 1. It consists of a box (1) where on its top was inserted a polystyrene integrating sphere (2) with 160 mm diameter and 50 mm thickness. The sphere was externally covered with an aluminum foil to remove external light noise. The inner part of the sphere is made of polystyrene and was assumed to have white-body-like properties such as an integral spectral signature that is independent from the light geometry. The fruit (3) is placed on the base of the IS and it is detained by a neoprene gasket with 60 mm diameter (4) between the sphere and a vacuum chamber (5). The vacuum chamber is a negative pressure space that has the function of sucking the fruit to obtain a complete adherence to the gasket, thus avoiding the possibility of having photons that escape and reach the sphere and the detector without passing through the fruit core. The chamber is made of a connecting pipe (6), which connects the vacuum chamber to a vacuum pump (7), in turn connected to the outside of the box through a gasket (8). The fruit is illuminated by a NIR 40 W light source (9) placed at the base of the vacuum chamber. The NIR light source spectral range start from 770 nm to 920 nm. Spectral data were collected by an Ocean Optics USB2000 spectrometer (Ocean Insight, Rochester, NY, USA) with a spectral sampling interval ranging from 350 to 1000 nm and a 0.2 nm spectral resolution, and internally based on a Sony ILX511 linear silicon CCD array (11, 12). The apparatus was controlled by an industrial PC and data were collected by Ocean View software (Ocean Insight, Rochester, NY, USA) (13). Light source and all the devices were powered by a 12V power supply (10). At the beginning and at the end of every acquisition round, a white Delrin^TM^ sphere, 80 mm diameter, was placed over the neoprene gasket in the same way as fruits, to collect reference transmittance spectra to be used to derive transmittances for all the measurements of that round. The use of such a reference sphere was necessary to obtain a reference spectra comparable with the magnitude of fruit spectra without changing the exposure interval of the spectroradiometer.

### 2.2. Experimental Measurements

The experiment was made at CNR (National Research Council of Italy) facility in Follonica (Italy) in April 2021. Seventy apples cv Golden Delicious at commercial maturity were collected from the mass retail channels and were stored in two plastic boxes with a size of 50 cm × 40 cm, thirty-five apples per box. Apples were inoculated with *Alternaria alternata* spp., one of the most representative fungi responsible for MC disease according to the replication and inoculation methodology already used by Ntasiu et al. [8]. The *Alternaria* was sampled by cv Golden Delicious in Valsugana (Italy), and it was characterized and preserved by Edmund Mach Foundation of San Michele All’Adige (Italy). The culture of pathogens was flooded with 5 mL of sterile distilled water and the conidia were scraped off with a surgical blade. The resulting conidial suspension was filtered through two layers of cheesecloth to remove mycelial fragments. Prior to inoculation, apple fruit surface was disinfected for 5 min by drenching them in a 1% NaOCl solution. The fruit were artificially inoculated by aseptically injecting 100 ul of a conidial suspension through the calyx into the fruit core with a syringe. Then, boxes were stored in a climatic chamber at 26 °C and 40 mL of water was added to each box that was covered to maintain relative humidity over 70%. Each sample was measured with the ALT-S every 3 days for 5 times for the total duration of the experiment (Table 1).

The operational analysis time (positioning of the fruit in the sphere and spectral acquisition) was approximately 90 s for each sample. Apples were sampled every time in two different positions, vertically (T1) and horizontally (T2) with respect to the NIR light source location. A total amount of 700 spectra signatures was collected in this way (70 samples × 5 times × 2 positions). After the final spectral acquisition, all fruits have been cut to check the growth of MC and the degree of its development. Fruits were positioned in an image acquisition platform to acquire RGB images to retrieve information about the amount of rotten versus healthy surface. The rotten area expressed as a percentage of the total cut area was retrieved with a threshold-based segmentation method operated in MatlabR2021a (MahtWorks, Natick, MA, USA) using Image Processing Toolbox. Biometrical data such as weight, volume, height, maximum and minimum diameter at the beginning and end of the experiment were collected. The MC data consisted of 70 values determined at the end of the experiment.

### 2.3. Data Analysis

#### 2.3.1. Transmittance Retrieval

The entire spectral dataset was composed of 700 spectral signatures of 2048 wavelengths (WL) ranging from 350 to 1000 nm. Only the spectral wavelengths within the range of the NIR light source were selected for the analysis, resulting in 750 bands from 770 nm to 920 nm. The set of 70 fruits was sampled in T1 and T2 positions for 5 measurement rounds. The spectral data were collected in absolute irradiance (uW/nm/cm^2^) and the spectral transmittance (TR) was computed as the ratio:TR = FRad/RRad(1)
where FRad is the radiance obtained by the photons transmitted through the fruit and RRad the radiance obtained by the photons transmitted through the Delrin sphere reference target.

#### 2.3.2. Band Ratios and Average Transmittance

A preliminary analysis was made to explore the acquired spectral dataset and verify the existence of a significant relation between MC measured at the end of the experiment and the spectral data measured at the 5-time steps along the experiment duration. The presence of such a relation can be considered as a prerequisite for the application of more complex machine learning classification methods, and may serve as an indirect estimate of the timing of the MC development by exploring different time steps. A simple band ratio index was computed for all possible combinations of couples of spectral bands transmittance ranging between 800 and 880 nm, separately for each fruit, each position, and each time step. Then the correlation coefficient between MC data (70 samples) and the multiple band ratios for the 70 fruits was computed for each measurement round and position, obtaining a correlation map for each round.

Furthermore, the average transmittance was computed for all the fruits belonging to the two classification categories (healthy and moldy), for each round, to derive additional qualitative indicators of the presence of spectral features associated to moldy state.

#### 2.3.3. Binary Classification

The overall objective of this study was to develop and test different binary classification models. The label-encoding was adopted assigning label 1 to moldy samples in relationship with the MC presence and label 0 to healthy samples. The spectral dataset was divided in one training dataset and four test datasets, where round 5 was used to train the model and rounds 1, 2, 3, and 4 were used as test. In all the classification models developed here, each WL represented an independent variable (IV) while the MC state represented the target variable to be predicted. Different binary classifiers were evaluated by computing performance metrics on accuracy, precision, and recall commonly used in machine learning models [39]. The performance metric was estimated by computing the confusion matrix on the training dataset [40]. By the definition of the confusion matrix, C is such that Ci,j is equal to the number of observations known to be in group i and predicted to be in group j. Thus, in binary classification, the count of true negatives is C0,0, false negatives is C1,0, true positives is C1,1 and false positives is C0,1. Therefore, the performance metrics were obtained to evaluate the best classifiers and to compare them.

##### Supervised Classification Models

Multiple supervised classification models (MSCM) based on the scikit-learn python library [41] were evaluated in this study; respectively: LR, SVC, RF, NB, KNN, and BC [42,43]. The model parameters of the RF, NB, KNN, and BC were optimized by cross-validated grid-search over a parameter grid [44]. The best performing model was selected, and then further improved by a dimensionality reduction aimed at reducing the number of independent variables. In fact, the presence of redundant information in the spectral data can distort classification results [45]. In classification problems, some statistical techniques can be used to minimize redundant data [46]. Here, we applied a univariate feature selection, as univariate statistical test to select k features that have the strongest relationship with the output variable. To select a specific k number of features, the ANOVA F-value method [47] via the Sklearn f_classif() function was used, while a grid search was implemented for the tuning of k [48].

##### Pattern Recognition Neural Network

A binary classification model based on ANN-AP was developed using MatlabR2021a Pattern Recognition Toolbox (MathWorks, Natick, MA, USA). Backpropagation with Momentum Algorithm (BMA) represents a powerful tool to resolve non-linear problems and it was selected to train the network [49,50]. BMA is a particular class of backpropagation algorithms where the input units are propagated forward to the output layer through the connecting weights. An accurate description of BMA can be found in the work of Phansalkar et al. [51]. The network’s architecture was developed in accordance with backpropagation rules and it was formed by 252 input, one layer with 252 hidden layers, one layer with two hidden layers, and two output labels. The train function ‘traingdm’ based on Fletcher-Powell Conjugate Gradient was used. The limit of training periods was set at 600 epochs. Other settings have been set at their default values.

## 3. Results and Discussion

### 3.1. Infection Rate

The infection rate was computed at the end of round 5 by destructive samplings and image segmentation. Results showed an infection rate of 54.2% (38 fruits over 70) against 45.8% of samples that remained healthy (33 fruits over 70). Based on the segmentation results, all the 70 samples were classified by the infection rate (Figure 2).

A relevant fraction of the infected samples (44%) had an infection level between 1% and 2% (Figure 2c). This condition made it possible to select an infection threshold of 0.51% to balance the number of samples labelled as healthy and moldy (respectively, 35 moldy and 35 healthy samples) in the binary classification training dataset. The choice of the infection threshold represents a critical issue in MC early detection [52] and has economic and industrial implications. The selection of a relatively large threshold on the one hand would facilitate the development of good classification algorithms, but on the other hand would carry the risk to classify fruits with infection rates lower than the threshold as healthy, which has relevant negative industrial impact [53]. The threshold selected here is remarkably low, improving over ten times the minimum infection level used in a similar previous study [35]. Keeping the minimum detectable infection at a low value represents a primary challenge for the performance assessment of the machine learning based methods presented here.

### 3.2. Spectral Correlations

The maps of correlation between transmittance band ratios and MC across all possible permutations of couples of bands was computed in T1 and T2 fruit positions and for each time step (Figure 3).

These maps show the absence of any significant correlation at round 1 and 2 in both T1 and T2 positions, with values contained within −0.2 to 0.2 not associated with any consistent pattern. This result is likely related to the absence of infection at these early stages of the experiment. At round 3, correlations are still very low for T1 position, while they reach values of 0.43 in T2 position, likely indicating that spectral proxyes of the infection progress started to be detectable at this stage. The presence of a distinctive higher correlation region was then observed at round 4 and then round 5, with maximum correlation values in T1 position of 0.67 and 0.82 and 0.74 and 0.51 in T2 position. This evidence supports the hypothesis that MC is developing within the fruits between round 1 and round 3, it is partially developed and detectable at round 4 and fully developed at round 5, when the destructive sampling and MC determination were made. The spectral bands whose ratio was associated with the maximum correlation are very similar in round 4 and 5, at 850 nm and 805 nm in round 4, and 849 nm and 802 nm in round 5.

Correlations between band ratios and MC in T1 position (vertical) are remarkably higher than in T2 position (horizontal), revealing that the measurement position influences the MC detection capability by affecting the spectral geometry. The explanation for this difference is likely related to the geometric symmetry of apple fruits along the vertical axis that reflects in anisotropic conditions and photons homogeneously passing through the internal parts of the fruit. On the other hand, the fruit placed in a horizontal position is not symmetrical along the vertical axis aligned with the light source, likely generating fruit specific geometric conditions affecting the light penetration and the spectral sampling. Given this difference in the performance, only the spectra retrieved in T1 position were selected for the subsequent analysis and machine learning classification.

### 3.3. Transmittance Temporal Pattern

The average spectral transmittance pattern in T1 position was obtained for each round (Figure 4).

In round 1 and 2, the TR curve does not show any significant difference between healthy and moldy samples. In round 3, the magnitude of TR from 860 nm to 880 nm increased in infected samples more than in healthy ones. In round 4, the difference in TR between healthy and moldy samples is further accentuated, especially from 800 to 820 nm and from 850 to 880 nm. In round 5, results show the maximum TR difference between healthy and moldy samples. The standard deviation of the infected population increases significantly compared to healthy samples. In addition, round 5 shows an increase of infected population’s transmittance from 825 to 880 nm. Similar studies were conducted on Fuji apples by Tian et al., and Zhaoyong et al. [34,36]. Tian et al. measured in moldy samples an increase of TR in the spectral region from 775 to 830 nm and a decrease from 830 to 880 nm. In contrast, Zhaoyong et al. showed a decrease of TR in moldy samples from 700 to 820 nm. Generally, the transmittance temporal pattern obtained in our work shows a decrease of TR in moldy samples from 800 to 830 nm and an increase from 830 to 880 nm, in contrast to Tian et al.’s results and in accord to Zhaoyong et al.’s results. The difference in results might be due to the different apple varieties used in previous spectroscopy analysis. Indeed, apple varieties present morphological differences such as their texture, skin color, and chemical composition and the spectral response results are inevitably influenced by these characteristics [54]. In addition, TR differences encountered in several studies might also depend on the large number of pathogens related to MC disease [7]; therefore, a precise taxonomic classification of the detected pathogens would be recommended.

### 3.4. Binary Classification and ANN-AP

#### 3.4.1. Features Reduction

Based on ANOVA univariate results, the F-value was computed in round 5 to select a specific k number of features to reduce the redundant information (Figure 5).

Results show a constantly increasing pattern of F-value from 830 nm to 866 nm, followed by a decrease and then a remarkable increase from 870 to 878 nm. Figure 5 shows two peaks in the spectral region from 860 to 878 nm caused by a larger variance in the spectral region from 860 to 880 nm. The F-value increases significantly from 860 to 878 nm in accordance with the increase of TR (Figure 4) obtained in temporal pattern results to the same spectral range and round. Therefore, spectral bands characterized by larger F-value indicate the spectral features most influenced by MC presence. Results obtained by ANOVA analysis allowed the reduction of the spectral range in MSCM training from 863.38 to 877.69 nm.

#### 3.4.2. Binary Classification

Several MSCM and one ANN-AP were assessed in this study. Between MSCM, the BC reported the best training score in accuracy and precision; consequently, classification results obtained from BC and ANN-AP were compared. Training results are shown in Table 2.

ANN-AP reported a higher level of precision (0.89) while BC model reported a higher level of accuracy (0.95) and recall (0.88). The accuracy measures the number of correct predictions made divided by the total number of predictions made. The precision is the number of true positives divided by the number of true positives and false positives. The recall is the ability to find all relevant instances (intuitively the ability of the classifier to find all the positive samples). The higher level of accuracy and recall reported by the BC model shows a better ability to classify correctly samples labeled (both positives and negatives) in comparison to the ANN-AP model. However, the higher level of precision reported by ANN-AP suggests a better ability to find positive instances. Indeed, a high score of precision reflects a high degree of discrimination between positive and false positive samples in training set. The precision score obtained in the training set confirms the higher capability to classify a low number of false positive by ANN-AP, as already reported in literature concerning the hyperspectral classification based on decision tree and neural network [55,56]. The importance of high precision in the training set is accentuated in a quality control industrial context, where the identification of infected samples has the greatest impact compared to the elimination of healthy samples. Therefore, failures in correctly detecting infected samples leads to a direct damage for the end consumers, which require healthy and standardized products. Differences in classification results by ANN-AP and BC models can be due to their different processing of the input variable. Indeed, ANN-AP is an algorithm inspired by biological neural networks, instead BC is based on a top-down approach of looking at the data. In ANN-AP, the value of the weights selected during the training process and the training goal is to minimize the error between values predicted by ANN-AP and true values [57]. The BC model uses a binary tree graph to assign for each data sample a target value and the target values are presented in the tree leaves. To reach the leaf the sample is propagated through nodes [58].

The two models, calibrated in round 5, were then tested on the previous rounds to assess them on completely independent datasets, also characterized by a different (e.g., lower) level of MC compared to the level used to train the models. Classification results on previous rounds are reported in Figure 6.

Overall, results obtained from both the BC and ANN-AP models exhibit an increase over time of the amount of detected infected samples; this behavior is in agreement with an expected exponential development of the infection [59]. The lowest number of positive samples was detected in round 1, respectively, 10% in BC model and 0% in ANN-AP. At round 1, a condition of complete absence of the infection was likely present, given that the inoculus was just applied at the beginning of the experiment. This condition is only met with the ANN-AP model, while the BC model reported a relatively large number of false positives (seven samples classified as positive). In round 2, the BC and the ANN-AP models detected an amount of 35.72% and 2.85% of positive samples, respectively. The temporal proximity of round 2 from to the inoculation time (six days) discourages the hypothesis that a high level of infection was reached. The correlation maps (Figure 3) and the average transmittance patterns (Figure 4) also support the hypothesis of a complete absence of infection at round 2. Therefore, BC model is likely reporting a large number of false positives while ANN-AP is performing properly. This difference can be attributed to ANN-AP learning skills that better adapt to new data patterns and reveal a higher capacity to interpret non-linear problems, as reported in the literature from Rojas and Abiodun et al. [60,61]. In round 3, the percentage of detected positive samples resulted similarly in the two models, at 40 % and 44% for BC and ANN-AP, respectively.

Both models suggest the presence of the infection at round 3, a finding that is supported by the correlation maps (Figure 3) and transmittance patterns (Figure 4), which reveal the first signs of the MC presence at this stage. It is possible that the machine learning methods have a higher specificity compared to a simple band ratio or an average transmittance pattern and detect an actual starting of the infection more effectively. Nevertheless, the possibility of these methods reporting false positives at this stage cannot be ruled out, and further research is needed to verify the actual infection rate at an early stage by destructive samplings that could be done only at the end of the experiment.

In round 4, 48.57% and 64.29% of samples were classified as positives in the BC and ANN-AP models, respectively; a large spread of infected samples at this stage is also confirmed by the correlation maps (Figure 3) and the average transmittance patterns that are significantly different between infected and healthy samples (Figure 5). Based on MC presence in round 5, the ANN-AP shows an overestimation of positive classifications. Overall, the classification models trained in this study might exhibit a possible uncertainty due to the limited size of the dataset [62].

This analysis gives insight on the model capability to detect MC at different development stages and it provides information about the model general applicability in real post-harvest condition. As already discussed, in an industrial context, the economic damage for primary producers due to the inputs of the infected samples in large scale retail trade is greater than the damage associated to the healthy product discard.

The BC model, while having an overall good classification capacity on the training dataset (Table 2), reported a large fraction of false positives at previous time steps while used in testing mode. This characteristic prevents its application in industrial detection given that the amount of infection and the temporal stage of the infection process are generally unknown. The ANN-AP model, on the other hand, exhibited both a good precision on the training dataset, and a consistent reproduction of the infection rates and development on the testing datasets, therefore, resulting in having a greater potential for industrial applicability. This potential should be further confirmed and assessed by means of further research deploying a significantly larger number of sampled and inoculated fruits, also encompassing infection variability across different cultivars.

## 4. Conclusions

This study proposes a novel early detection system based on NIRS technologies to detect MC in cv Golden Delicious apples. The measurement device, called the ALT-System, was successfully developed and tested on a set of infected samples. The ALT-System has demonstrated its capability to detect MC through one single measurement per fruit, surpassing the previous measurement systems that required repeated measurements. Therefore, the achievement of this goal represents an important starting point to develop an efficient industrial prototype. In this study, spectral features linked to the MC presence were identified in the spectral region from 863.38 to 877.69 nm, and the spectral differences in term of TR between healthy and infected samples were explained over five different measurement rounds during the development of the infection. Several binary classification methods based on decision trees and ANN methods were tested here. ANN-AP and BC models were the two best methods with a better score in terms of performance metrics such as accuracy, precision, and recall. Moreover, their performance in training and validation datasets were assessed and discussed. By comparing training results between the two methods, the ANN-AP training results exhibited the best score in precision (0.89), a parameter whose optimization is mostly associated with the industrial needs of fruits post-harvest processing. Indeed, an efficient capability to distinguish healthy samples is a primary requirement to develop a functional industrial detection system. BC and ANN-AP models were tested on independent datasets obtained at previous stages of the infection development, and the ANN-AP showed a better classification result in relation to the infection growth rate. Indeed, ANN-AP showed a predictive accuracy of 100% at round 1 and 97.15% at round 2, demonstrating a better capability compared to BC model to interpret unknown datasets. Classification errors reported here might be related to the small dataset size and a low infection threshold adopted.

In an industrial context, MC early detection through non-invasive methods remains a critical issue, and this work represents a starting point to develop an industrial-scale prototype based on NIRS. Further studies are recommended to develop a measurement system capable to overcome the limits of the ALT-System: the inability to measure multiple fruits simultaneously and the need to minimize the measurement time per fruit. In addition, further investigations are needed to integrate the ALT-System directly on a conveyor belt. Additional spectral tests on multiple apple varieties are also recommended to explore the link between skin color, texture, biochemical contents, and spectral response. Finally, the technology and methods used and developed in this work might be applied in other apple diseases detection, e.g., biotic and abiotic internal browning or post-harvest CO_2_ damages, and show a high potential for application to other fruits.

## Figures and Tables

**Figure 1 sensors-22-04479-f001:**
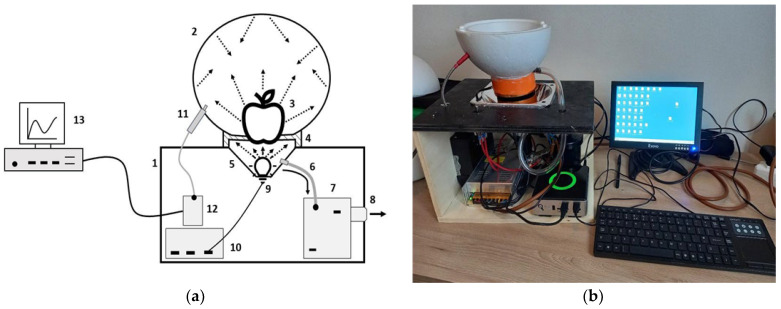
ALT-System (Apple Light Transmittance System) operation diagram (**a**) and ALT-System instrument (**b**).

**Figure 2 sensors-22-04479-f002:**
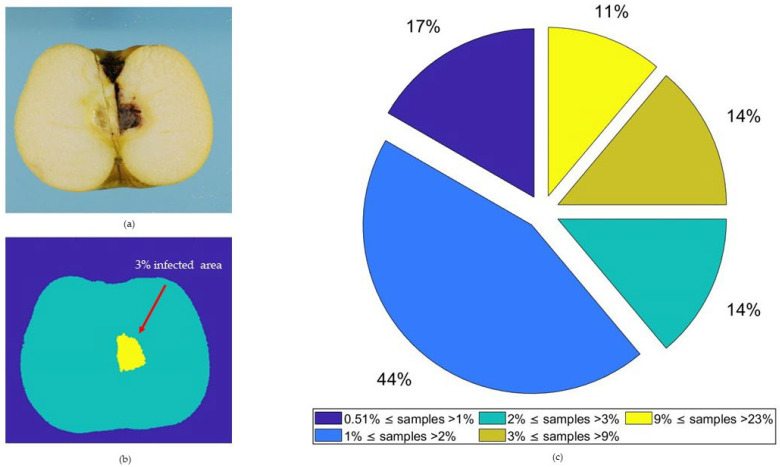
Example of segmentation results in a sample with a 3.0% infection rate. RGB image (**a**) was isolated from background, the results obtained (**b**) were furthermore processed to obtain necrotic area (yellow) separated from healthy area (light blue). Infection results were grouped into classes based on infection levels (**c**).

**Figure 3 sensors-22-04479-f003:**
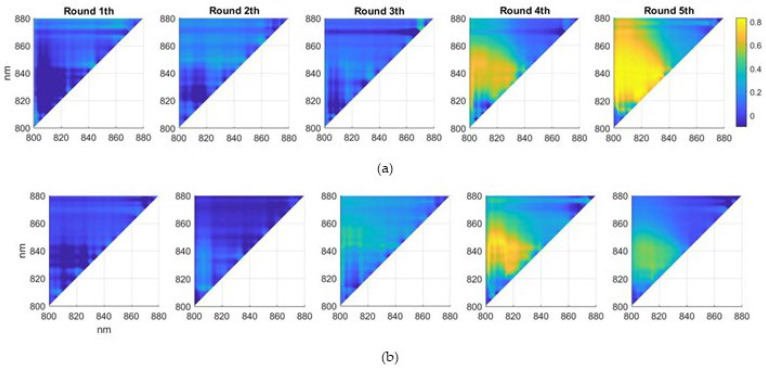
Maps of the correlation coefficient between MC measured at the end of the experiment (round 5), and transmittance ratios Tx/Ty based on two individual bands, computed on all combinations of x and y ranging over the sampled spectral interval (800–880 nm) for each round (1 to 5). (**a**,**b**) refer to T1 and T2 positions, respectively.

**Figure 4 sensors-22-04479-f004:**
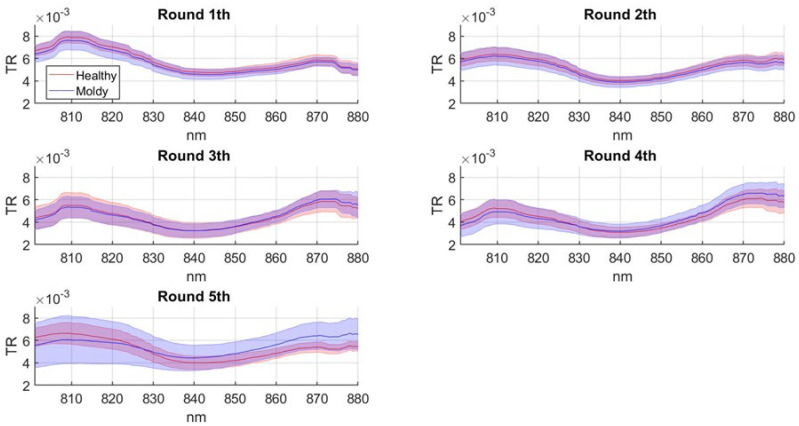
Temporal pattern of mean transmittance for healthy and infected samples with a threshold infection level of 2% or more. Transmittance was first averaged across all wavelengths and then mean and standard deviation were computed.

**Figure 5 sensors-22-04479-f005:**
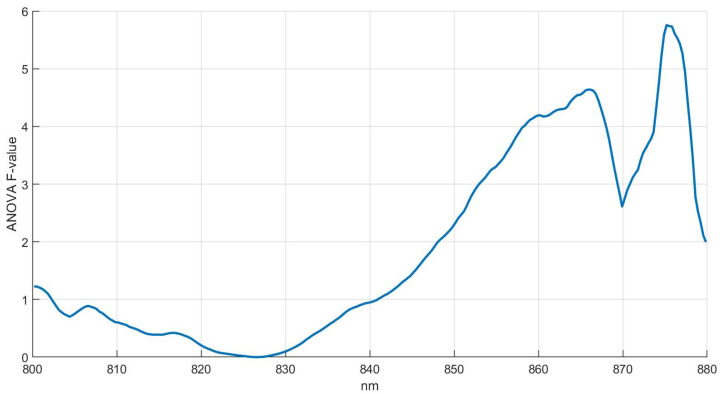
ANOVA F-value results from 800 to 880 nm. F-value is a ratio between two variances and a higher F-value corresponds to a significant statistical mean separation between healthy and moldy groups.

**Figure 6 sensors-22-04479-f006:**
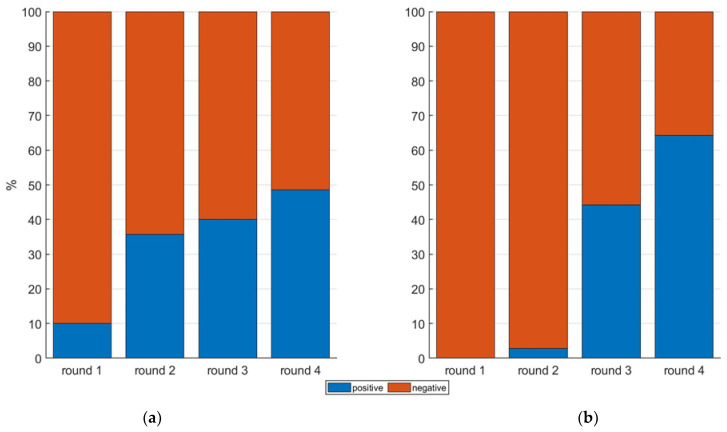
BC model (**a**) and ANN-AP (**b**) test results in round 1, 2, 3, 4. The binary classification results on unlabeled datasets show two classes, respectively, infected (positive) and healthy (negative). The two models were trained on round 5 with a threshold infection rate of 0.51%; therefore, the number of positive samples and negative in the training set resulted in 35 and 35 (50% positive and 50% negative), respectively.

**Table 1 sensors-22-04479-t001:** Experiment timetable.

Days	Round	Operations
2 April 2021	1	Biometrical measurement, inoculation, and spectral acquisition
5 April 2021	2	Spectral acquisition
8 April 2021	3	Spectral acquisition
11 April 2021	4	Spectral acquisition
14 April 2021	5	Spectral acquisition, biometrical measurement, MC presence validation, and RGB acquisition

**Table 2 sensors-22-04479-t002:** BC model and ANN-AP training results. The binary classification models were trained on round 5 and the performance metrics (accuracy, precision, and recall) were computed by confusion matrix.

Model	Accuracy	Precision	Recall
BC	0.95	0.85	0.88
ANN-AP	0.72	0.89	0.62

## Data Availability

Not applicable.

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
