# Peer review of "A Novel Hyperspectral Method to Detect Moldy Core in Apple Fruits"

_sensors, 2022, doi:10.3390/s22124479_

Round 1

Reviewer 1 Report

In this paper, authors described and proposed a novel hyperspectral approach for non-invasive internal browning detection of apple core. Their principle and thoughts are well established and look thoughtful. However, this paper can be recommended for publication after minor revisions, namely,
1) line 41: 'Alternaria sp (Asp)' should be highlighted as italic. Please, check in the whole manuscript.
2) lines 96 and 341: 'Error! Reference source not found.' Please, correct references in the whole manuscript.
3) lines 44-49: why do Golden Delicious apple variety is so special than other species?
4) line 90: word 'cv' was mentioned without definition. Please, define it.
5) lines 123-124: you mentioned about industrial implementation. Please, describe how can you scale your setup for industry? Is it possible to detect moldy core if apples will be supplied in boxes?
6) How fast is your method?

Author Response

We thank the Reviewer for the positive appreciation of our work. Please find below replies to each comment in red.

1) line 41: 'Alternaria sp (Asp)' should be highlighted as italic. Please, check in the whole manuscript.  Corrected
2) lines 96 and 341: 'Error! Reference source not found.' Please, correct references in the whole manuscript. Corrected
3) lines 44-49: why do Golden Delicious apple variety is so special than other species? Golden Delicius is the most widely cultivated apple cv in Italy, with a total production volume of 858423 tonnes. We added this information to the text (lines 95-98)

4) line 90: word 'cv' was mentioned without definition. Please, define it. Cultivar (cv) definition is now reported (line 93).
5) lines 123-124: you mentioned about industrial implementation. Please, describe how can you scale your setup for industry? Is it possible to detect moldy core if apples will be supplied in boxes?

- The final objective of this work will be to develop a scale prototype based on the model proposed in this article. However, we need to optimize few issues, e.g:

  1. develop a system capable of measuring multiple fruits simultaneously.
  2. Minimize the measurement time per fruit.
  3. Integrate the measurement system in a quality-control line.

These potential developments were added in section 4 “conclusion” (lines 760-786).

The developed system only measures one fruit at a time. Therefore, the apples can’t be analysed on boxes.

6) How fast is your method? The time it takes in the current setup to process a single apple (e.g placing the apple in the sample position, taking the spectral measurement and removing it from the position) is about 90 seconds. We added this information to the text (line 373-374). Of course, this time must be drastically reduced if industrial applications are to be designed capable of scaling the system architecture.

Reviewer 2 Report

This study presents a unique early detection approach for MC in Golden Delicious apples based on NIRS technology. The ALT-System measuring device was created and tested successfully on a batch of contaminated samples. The ALT-System has shown that it can identify MC with just one measurement per fruit, compared to earlier measuring methods that required many measurements. The paper is interesting and could be published in the Journal of sensors in the reviewer's view if the following comments and other reviewers, comments to be fulfilled.

- Abstract could be more informative by providing results. I prefer to see some results in the abstract.

- it is always expected to use the following structure when writing your abstract: (a) Background or Introduction, (b) Aim/Goal, (c) Experimental Method/Approach, (d) Results and Discussion, and (e) Scientific Impact. The abstract should be a precise and short summary of the whole manuscript. The authors focused more on few of the (previous mentioned) points. Please improve/rewrite it.

- Please use some innovative keywords.

- Please mention your study limits in the abstract.

- The introduction needs to be more emphasized on the research work with a detailed explanation of the whole process considering past, present and future scope. How the present study gives more accurate results than previous studies? It needs to be strengthened in terms of recent research in this area with possible research gaps. It is strongly recommended to add a recent literature.

- Selected references are quite old, which from the one point of view is good, since the authors cited necessary references to define a research problem, while from the other hand, lack of recent references may indicate an insufficiently performed literature review. Try to refer to some recent and up-to-date research papers related to the topic, especially in recent years like those related to Artificial Intelligence and Artificial Neural Network:

- A novel approach to predict shear strength of tilted angle connectors using artificial intelligence techniques

- The authors have to explain what is the new here in comparison with the previous studies.

- The novelty of the current work should be highlighted in the introduction

- Please try to mention a problem that needs solving - in other words, the research question underlying your study more clear.

- Please kindly make revision on the language of the paper presentation. There are still some minor typos and grammatical errors.

- Please improve the quality of pictures in the manuscript.

- In section 1, after the State of the Art (SoA), the Authors should clarify what are the key novelties of this paper and the main contributions of this work beyond the current SoA. They are barely addressed.

- Try to omit the use of he, we. Use passive voice instead.

- Check for format errors (e.g., extra spaces, use of "_")

- How did you collect reference transmittance spectra to be used to derive transmittances?

- Please report enough detail in order to replicate the study.

- Plots of all Figures need to be uniformed in size and style

- Did you repeat the experimental tests to assure the results?

- How many tests did you do in this study?

- Please explain test set up in more detail.

- How many times did you measure each sample during the day?

- Did you measure the Biometrical data of each sample?

- Explain figure 3 in more detail.

- Does the color of apples (Red or yellow) affect the results?

- How did you make difference between inflected and healthy samples in your model? What was your parameter?

- Explain the difference between ANN-AP and BC model.

- Please compare your results with some papers in the related field and explain the similarity and differences between your results and theirs.

- The discussion of the comparison results may be strongly extended, by providing proper considerations to each plotted graph

Major Comment: The conclusion should be an objective summary of the most important findings in response to the specific research question or hypothesis. A good conclusion states the principle topic, key arguments and counterpoint, and might suggest future research. It is important to understand the methodological robustness of your study design and report your findings accordingly. Please improve your conclusion section.

- Please describe the scope of your research and make recommendations for future research subjects.

- The authors should provide a complete conclusion that includes essential values, the applicability of the applied approach, contributions, and prospective future work.

- By giving adequate thoughts to each displayed graph, the discussion of the comparative findings Section may be greatly expanded.

Reviewer 3 Report

The authors describe an interesting application to detect moldy cores in apples usin NIR spectroscopy.

The following revisions should be considered:

Line 20: check nomenclature of organisms. The second word should not be upper case.

Line 34 and throughout: check style for in-text references (not bold)

Introduction, last two paragraphs: revise, they more or less state the same information twice (objectives).

Section 2.2.: how long does an experiment take? Would this be possible in industrial practice on conveyor belts?

Line 341: check error in reference source

Round 2

Reviewer 2 Report

The revised manuscript could be accepted for publication after my checking.